# Microencapsulated and Lyophilized Propolis Co-Product Extract as Antioxidant Synthetic Replacer on Traditional Brazilian Starch Biscuit

**DOI:** 10.3390/molecules26216400

**Published:** 2021-10-22

**Authors:** Rodrigo Rodrigues, Denise Bilibio, Manuel S. V. Plata-Oviedo, Edimir A. Pereira, Marina L. Mitterer-Daltoé, Ellen C. Perin, Solange T. Carpes

**Affiliations:** 1Department of Chemistry, Federal Technological University of Paraná (UTFPR), Pato Branco 85503-390, Brazil; rrodrigo.rdr@gmail.com (R.R.); edimir@utfpr.edu.br (E.A.P.); marinadaltoe@utfpr.edu.br (M.L.M.-D.); ellenperin@utfpr.edu.br (E.C.P.); 2Federal Institute of Rio Grande do Sul-IFRS, Sertão 99170-000, Brazil; denise.bilibio@sertao.ifrs.edu.br; 3Department of Food Technology, Federal Technological University of Paraná (UTFPR), Campo Mourão 87301-899, Brazil; mapaov@utfpr.edu.br

**Keywords:** ethanolic extract of propolis co-product, spray drying, polvilho biscuit, fatty acids, antioxidant of lipid oxidation

## Abstract

The residue from commercial propolis extraction may have significant antioxidant power in food technology. However, among the challenges for using the propolis co-product as an inhibitor of lipid oxidation (LO) in baked goods is maintaining its bioactive compounds. Therefore, this study aimed to determine the propolis co-product extracts’ capability to reduce LO in starch biscuit formulated with canola oil and stored for 45 days at 25 °C. Two co-product extracts were prepared: microencapsulated propolis co-product (MECP) (with maltodextrin) and lyophilized propolis co-product (LFCP), which were subjected to analysis of their total phenolic content and antioxidant activity (AA). Relevant antioxidant activity was observed using the methods of analysis employed. The spray-drying microencapsulation process showed an efficiency of 63%. The LO in the biscuits was determined by the thiobarbituric acid reactive substances (TBARS) test and fatty acid composition by gas chromatography analysis. Palmitic, stearic, oleic, linoelaidic, linoleic, and α-linolenic acids were found in biscuits at constant concentrations throughout the storage period. In addition, there was a reduction in malondialdehyde values with the addition of both propolis co-product extracts. Therefore, the propolis co-product extracts could be utilized as a natural antioxidant to reduce lipid oxidation in fatty starch biscuit.

## 1. Introduction

Bees extract their necessary nutrients from plant sources. These sources contain various compounds, including compounds related to specialized metabolism, especially phenolic compounds. Due to the presence of these compounds, propolis has been characterized as having properties associated with health benefits, for example, antioxidant, antimicrobial, anticoagulant, healing, and anti-inflammatory activities, among others [1,2,3,4,5].

However, to obtain the propolis extract for sale, a residue is generated, hereinafter called a co-product. Thus, finding ways to achieve improved economic use, and the possibility of applying the co-product as a natural antioxidant in food technology, has been of great interest [6,7,8,9]. However, in this effort to exploit the extract, an important factor, considered as one of the main challenges, is the maintenance of compounds with antioxidant potential and their degradation during food processing, since they are susceptible to the effects of various conditions, such as exposure to moisture, light, and oxygen [10]. Furthermore, acceptance of the characteristic aftertaste of propolis has become a challenge for application [1,4,7].

A possible approach to minimizing the taste and odor effects of these compounds is microencapsulation. Microencapsulation can be defined as a process in which gaseous, solid, or liquid materials are wrapped in a material as a coating to form micrometer-by-millimeter capsules. Different techniques can be used, but they are essentially classified into three methods: chemical, physical, and physicochemical methods, with differing advantages and disadvantages for each method [11].

Microencapsulation, with an appropriate carrier, is an alternative technology that can serve to mask the astringent taste generated by phenolic compounds and achieve greater stability in food storage [1,4,11,12]. In this study, despite possible adverse sensory effects and potential losses of phenolic compounds by cooking, the use of the lyophilized extract of the propolis co-product was also considered in order to evaluate its effect in the inhibition of lipid oxidation.

This technique has been used in different food substrates, with a focus on bioavailability and the protection of bioactive compounds in products such as coffee [10], pomegranate [13], and blackberry [14], as well as in the development of functional foods [15], for extending the shelf-life of food products [16], and other purposes. Furthermore, applications of microencapsulation and encapsulation in propolis have been performed using several coating materials and methods [17,18,19,20,21,22], in different food systems, and with a focus on health benefits [23,24].

Due to the antioxidant potential previously established for propolis [6,8], the use and application of the co-product in food products susceptible to lipid oxidation would be of great value. The oxidation of lipids present in food directly affects the product’s quality and shelf life. It alters the composition, texture, and flavour of food, decreasing its nutritional value. Lipid oxidation reactions can cause severe damage to the food industry [1,12].

The use of synthetic or natural antioxidants in foods rich in oils and fats slows down or inhibits lipid oxidation, directly affecting the quality and acceptability of foods. In food industrialization, the use of synthetic antioxidants, such as butylated hydroxytoluene (BHT), butylated hydroxyanisole (BHA), and tert-butylhydroquinone (TBHQ) is common. However, these additives are cited for their harmful health effects and are banned in many countries [25,26].

The present study aimed to develop starch biscuit (polvilho biscuit) based on canola oil containing propolis co-product extracts (LFCP and MECP) and to evaluate the antioxidant potential of these extracts as a lipid oxidation inhibitor. Different formulations of traditional Brazilian starch biscuits, based on canola oil with LFCP, MECP, and BHT, were prepared to assess lipid oxidation during room temperature storage.

## 2. Materials and Methods

### 2.1. Chemicals

Fatty acid methyl ester (FAME) standards C4-C24, 2,2-diphenyl-1-picril-hydrazil (DPPH), fluorescein, 6-hydroxy-2,5,8,8-tetramethylchromo-2 acid-carboxylic (Trolox), 2,2-azino-bis-(3-ethylbenzothiazolin-6-sulphone acid) (ABTS+), and 2,4,6-Tris (2-pyridil)-s-triazine (TPTZ) were purchased from Sigma-Aldrich (St. Louis, MO, USA). Butyl hydroxytoluene (BHT), 2-thiobarbituric acid, and Folin-Ciocalteu phenol were purchased from Vetec (São Paulo, Brazil). Sour starch from General Mills (Paranavai, PR, Brazil), maltodextrin from Cargill Agrícola S.A. (São Paulo, SP, Brazil), and canola oil from Bunge Alimentos S/A (Gaspar, SC, Brazil) were acquired in local commercial outlets in Chapecó, Santa Catarina, Brazil.

### 2.2. Samples

A sample of the propolis co-product originating from propolis collected in apiaries of União da Vitória-PR during the 2018 Spring (latitude 26°11′48.8′′ S, longitude 51°06′48.4′′ W) was used. The propolis co-product, the residue resulting from the commercial propolis extraction process, was supplied by Breyer & Cia LTDA (União da Vitória, PR, Brazil). The residue was dried at room temperature for 48 h to achieve 7% humidity and was stored under refrigeration at −15 ± 0.6 °C.

### 2.3. Preparation of Propolis Co-Product Extracts

Aliquots of 10 g of the sample were extracted in 100 mL of ethanol (0.8 g mL^−1^) in an orbital incubator-type shaker (FORTINOX-Star FT38, Piracicaba, SP, Brazil) at 40 °C for 60 min and under agitation at 145 rpm. A total of 1000 mL of extract was prepared at a concentration of 0.1 g mL^−1^ and filtered on filter paper. The supernatant was used to obtain the LFCP and MECP by the same method as [1].

### 2.4. Lyophilized Propolis Co-Product (LFCP)

The propolis co-product extract’s supernatant (100 mL) was evaporated in a rotary evaporator (TE-211, TECNAL, Piracicaba, Brazil) until complete evaporation of the solvent. The residue was then lyophilized (Lyophilizer L101-LIOTOP, São Carlos, SP, Brazil) under a high vacuum for 48 h at a temperature of −50 °C. The lyophilized residue was stored under refrigeration at −18 °C for further analysis.

### 2.5. Preparation of Microencapsulated Propolis Co-Product (MECP)

Approximately 900 mL of the propolis co-product extract’s supernatant was concentrated in a rotary evaporator at 40 °C until a 50% reduction in its initial volume was achieved. A spray dryer microencapsulated the extract with a total solids content of 21.4% (Brix).

### 2.6. Emulsion

Before drying in the spray dryer, an emulsion was prepared with 50 g of maltodextrin from Cargill Agrícola S.A. (São Paulo, SP, Brazil), which was dispersed in 50 mL distilled water at room temperature under mechanical agitation at 2500 rpm (Fisatom 713D–AAKER, São Paulo, Brazil) until dissolution was complete. A further 50 mL of distilled water at 90 °C was then added, and agitation increased to 4300 rpm to obtain the emulsion. The emulsion was prepared consistent with previous reports by the team [1,27]. The concentrated extract of propolis was then added and homogenized for introduction into the spray dryer.

### 2.7. Drying Conditions

The spray dryer LM MSD 1.0 (Labmaq, Ribeirão Preto, SP, Brazil), with an atomizer for two fluids (air/sample) of a diameter of 1 mm, was used to obtain the microencapsulated material. The emulsion (encapsulant + extract) under constant agitation (1000 rpm) was pumped via an inflow into the air stream at a rate of 0.70 L h^−1^. The temperature of the inlet and outlet air was 150 °C and 72 ± 2 °C, respectively. The rate of entry of air entering the equipment (atomization) was 3.6 L min^−1^, while the compressor rate ranged from 45 to 50 L min^−1^.

### 2.8. Microencapsulation Efficiency Analysis (ME)

The efficiency of the microencapsulation process was determined according to previous studies by [1] and [27]. The difference between the percentages of the total coated phenolic compounds present inside the microcapsules (TPM) and the total phenolic compounds present on the surface (TPS) was calculated (Equation (1)). The test was carried out in triplicate.
ME (%) = [1 − (TPS/TPM) × 100](1)
where:

TPS: Total phenolics on microcapsule surface.

TPM: Total phenolics of microcapsule.

### 2.9. Scanning Electron Microscope (SEM)

The structural characteristics of the microencapsulated material (MECP) were analyzed by digital scanning electron microscopy (SEM) (TM-3000, Hitachi Tabletop Microscope, Tokyo, Japan). The sample (powder) was adhered to double-sided adhesive tape and arranged in a metal sample port (stubs) of 12 mm in diameter. The SEM images were obtained using a voltage of 5 kV and a current of 1750 mA. The software, Shadow 1 (Hitachi Tabletop), recorded the magnifications from 100 to 1500×.

### 2.10. Water Activity Analysis

The water activity (aw) of the materials (MECP) was measured by direct reading in the automatic analysis equipment (LabTouch Novosina–Lachen, Switzerland) according to the chemical-physical methods for food analysis described in [28], with triplicate analysis.

### 2.11. Total Phenolic Compounds

The content of total phenolic compounds was determined by [29]. Aliquots of 500 μL of LFCP and MECP at 0.2 mg mL^−1^ were solubilized with 2.5 mL Folin–Ciocalteu reagent (2 N) in Falcon tubes and kept at rest for 5 min. After the rest period, 2.0 mL of sodium carbonate solution (40 mg mL^−1^) were added and the solutions were homogenized. After this step, the tubes were kept for 2 h at room temperature without exposure to light. The absorbances were read in a spectrophotometer (Femto UV 2000, São Paulo, Brazil) at 740 nm.

### 2.12. Antioxidant Activity Analysis

#### 2.12.1. DPPH Radical Scavenging Assay

The antioxidant activity assessment was performed using the DPPH method according to [30]. The MECP and LFCP extracts were diluted in 80% ethanol and prepared at a concentration of 45.45 mg mL^−1^ and 0.2 mg mL^−1^, respectively. Aliquots of 500 μL of each extract were added separately to 3 mL of ethanol p.a. and to 300 μL of DPPH solution (0.5 mM). A control sample without extracts was prepared with 3.5 mL of ethanol and 300 μL of DPPH solution (0.5 mM). The samples were homogenized and kept protected from light for 30 min and absorbance readings were taken in a spectrophotometer. Extracts from the MECP and LFCP samples were analyzed in triplicate and the results expressed in μmol of Trolox g^−1^ sample.

#### 2.12.2. ABTS Radical Scavenging Assay

Radical ABTS (2,2’-azino-bis-(3-ethylbenzothiazoline-6-sulfonic acid) at a concentration of 7.0 mM was obtained in solution with potassium persulfate (140 mM) after 16 h of reaction at room temperature and without light incidence. A quantity of 3.0 mL of the solution containing the ABTS radical, and aliquots of 30 µL of each extract, MECP and LFCP, were diluted at concentrations of 50 mg mL^−1^ and 0.2 mg mL^−1^, respectively. The absorbance of the reaction medium containing ABTS was adjusted in a spectrophotometer up to 0.7 absorbance at a wavelength of 734 nm [31]. A standard curve was constructed with Trolox at concentrations ranging from 0.0 to 1500 μmol. The analyses were performed in triplicate, and the results expressed in antioxidant capacity equivalent to Trolox (μmol TEAC g^−1^).

#### 2.12.3. Ferric Reducing Antioxidant Power (FRAP)

In the analysis by the FRAP method [32], the FRAP reagent was prepared with 50 mL of acetate buffer (300 mM, pH 3.6), 5 mL of TPTZ solution (10 mmol of TPTZ in 250 mL of 40 mM HCl), and 5 mL of FeCl_3_ (20 mM) in aqueous solution. The MECP and FCP extracts were diluted at a concentration of 50 mg mL^−1^ and 0.2 mg mL^−1^, respectively; the analyzed aliquots contained 90 µL and were analyzed in triplicate, the calibration curve was considered with ferrous sulfate with a concentration between 100 to 2000 µM; the results obtained were expressed in μmol Fe^2+^ per g of sample (μmol Fe^2+^ g^−1^).

### 2.13. Brazilian Starch Biscuit Elaboration

LFCP and MECP extracts were applied in the Brazilian starch biscuit formulations to control the products’ lipid oxidation. The biscuit dough was prepared with the following ingredients: sour starch (411.20 g kg^−1^), salt (13.20 g kg^−1^), milk (164.50 g kg^−1^), canola oil (164.50 g kg^−1^), water (164.50 g kg^−1^), and eggs (82.25 g kg^−1^). First, the oil and water were heated to 100 °C and mixed with starch and salt (scalding). This scalded dough was homogenized for 3 to 5 min, and then a mixture of milk and eggs were added at room temperature. The dough was homogenized for a further 5 min and divided into four lots. The four formulations were: F1–control, no additional ingredient was included; F2–containing butylated hydroxytoluene (BHT) 100 mg kg^−1^ of fat; F3–containing MECP (500 mg kg^−1^ of fat); and F4–containing LFCP (750 mg kg^−1^ of fat). The research literature states that propolis and its co-product have previously been used as an antioxidant with concentrations ranging between 100 and 500 mg Kg^−1^ of food sample, in studies with meat products [1]. The amount of antioxidant considered in the starch cookies was randomly allocated, with the encapsulated extract (MECP) being used in a smaller amount than the lyophilized extract (LFCP). The cylindrical biscuits were moulded with bakery silicone pastry bags and were then baked in an oven (Fischer Fit-Line, Brusque, Brazil) for 12 ± 1 min at 180 °C. After cooling at ambient temperature, the biscuits were collected in polyethylene bags, sealed and kept at 25 ± 1 °C.

### 2.14. TBARS Assay

The oxidative stability of the starch biscuit formulations based on canola oil was evaluated by thiobarbituric acid reactive substances (TBARS) [27]. Measurements were made at 0, 15, 30, and 45 days of storage at 25 ± 1 °C. Oxidation was verified by the determination of malondialdehyde (MDA) concentration produced during the reaction. Aliquots of 2.5 g of the formulations were ground and dissolved in 25 mL of trichloroacetic acid and the sample was then filtered. A volume of 4 mL of the filtrate was added to 1 mL of 7.5% TCA (m.v) and 5 mL of 0.02 mol L^−1^ TBA. A calibration curve with the standard 1,1,3,3-tetramethoxypropane (TMP) was constructed using concentrations of 0.0 to 4.50 10^−8^ mols. The samples were analyzed in triplicate and the results expressed in mg of MDA kg^−1^ sample.

### 2.15. Fatty Acid Esters by Gas Chromatography

The total lipids extracted from the biscuits were submitted to esterification [33]. Obtaining the methyl esters of the fatty acids required submitting them to the reaction under heating with a solution of sodium hydroxide and methyl alcohol, then an esterifying solution (NH_4_Cl-H_2_SO_4_–MeOH) was added. The next steps consisted of phase separation and extraction with hexane. The fatty acid esters were determined by an Agilent chromatography system (7890B GC, Santa Clara, CA, USA), equipped with a flame ionization detector (FID) and an Agilent DB-Wax column (30 m × 0.25 mm × 0.15 μm). The oven temperature was programmed at the beginning to 50 °C for 2 min and increased up to 250 °C with a heating rate of 3 °C min^−1^. The injector’s temperature was set at 250 °C with 1 μL of each sample (split-50:1 mode) used on automatic injection. Hydrogen was used as a carrier gas at 40 cm s^−1^ flow. The total time for chromatographic analysis was 68.67 min. The fatty acids were quantified by the normalization method with the corrected area [34,35,36]. The samples’ lipid content (%) and the corresponding constitution of triacylglycerols in the composition of fatty acids in the vegetable oil samples was determined (Equation (2)). The analyses were performed in triplicate, and the results expressed as g of fatty acid per 100 g of biscuit (g 100 g^−1^ starch biscuit).
M (g 100 g^−1^ starch biscuit) = FCT × % by mass of FAME × TL(2)
where:

M: fatty acid mass

FCT: conversion factor for vegetable oils (0.956).

TL: decimal value of the lipid content of the sample.

FAME: fatty acid methyl ester standards.

### 2.16. Statistical Analysis

Statistica 8.0 software was used for the application of statistical tests. The data were evaluated to verify that they met required assumptions and submitted to variance analysis (*p* ≤ 0.05). The normality and homogeneity of variances were evaluated. The results were submitted to analysis of variance (ANOVA) and the Tukey test for comparison between means.

## 3. Results and Discussion

The physical-chemical characteristics, the identification and quantification of phenolic compounds, and the antioxidant potential of MECP and LFCP were evaluated for inclusion in Brazilian starch biscuit (polvilho biscuit) formulations.

### 3.1. Characterization of Lyophilized (LFCP) and Microencapsulated Co-Product Propolis (MECP)

The encapsulation efficiency was 63 ± 0.06% and corroborates previous studies of propolis microencapsulation with pea protein carriers made by [7], which obtained efficiency values of 49.4 to 97.0% at different concentrations of propolis extract. Other studies with microencapsulated propolis extracts showed higher efficiency values, such as those reported by [1,37] with values of 76.86% and 70.38%, respectively.

The encapsulation efficiency may be associated with factors such as the drying temperature of the spray dryer, chemical properties of the coating material and the material to be coated, as well as the preparation mode and emulsion dispersion (core/wrap) [1,7,37]. The propolis’ intrinsic characteristics, such as geographic origin, seasonality, and sample preparation, and extraction procedures may affect these differences [37,38].

The propolis co-product microparticles analyzed by SEM presented a spherical shape with few depressions in their structure, as shown in Figure 1. During the atomization process spheres were formed and the bioactive material was dispersed as fine solid particles within the matrix [39,40]. The microparticles should not have defects or pinholes to ensure a greater stability of coating material (maltodextrin) and thus, protect the bioactive component. The presence of defects may increase the rate of oxidation or hydrolytic degradation. The same behavior was observed for a soy protein isolate/tocopherol emulsion in a previous study by [40] that found microparticles had a spherical shape without defects.

The microcapsules showed a smooth and uniform appearance on the microcapsules’ surface, contributing to the microencapsulated materials’ fluidity [40]. This uniformity also contributed to the protected bioactive material avoiding exposure and degradation [7,41].

The result of the water activity analysis (Aw) of the extracts were less than 0.50, similar to the values cited by [1,7,37,41] and represent products with lower susceptibility to degradation. Table 1 presents the total phenolic content (TPC) results and antioxidant activity by three different methods (ABTS, DPPH, and FRAP). As expected, the lyophilized extract’s total phenolic compound content was approximately three times higher than the values found in the microencapsulated extract. In this study, the lyophilized extract showed significantly higher antioxidant activity (about ten times higher) than that found in the microencapsulated extract by the three methods tested.

In a previous study, [42] presented total phenolic content of 90.55, 55.74, and 91.32 mg GAE g^−1^ of green, brown, and red propolis, respectively, were obtained. The same samples had been encapsulated, showing a significant reduction in TPC values, which were not higher than 48.38 mg GAE g^−1^ [37]. These total phenolic compound contents were very close to those obtained with the LFCP and MECP extracts.

In extracts dried in the spray dryer in propolis samples with a wide mass range (5 to 30%-m/m) and solvent concentration (ethanol, 60 to 96% (*v*/*v*)), levels of phenolic compounds of 48 to 87 mg GAE g^−1^ were obtained [39]. The values were within the content range of total phenolic compounds established by [43] (31 to 299 mg GAE g^−1^) in propolis from several geographical sources.

Previous studies with crude propolis [38,42,44], a co-product of propolis [1,3], and microencapsulated material [1,7,12] demonstrated a very varied range of antioxidant activity values.

The DPPH radical scavenging method’s antioxidant activity was lower than that found for the ABTS scavenging test (Table 1). This behaviour is similar to the results obtained by [45] in organic propolis samples (290 to 1240 μmol Trolox g^−1^) by the ABTS method and of 10 to 380 μmol Trolox g^−1^ by the DPPH method).

The values obtained by the iron-reducing power method (FRAP) with the LFCP extract were also higher than those found in the MECP. These values correspond to the results obtained by [37] in propolis samples (green, brown, and red) encapsulated with arabic gum and maltodextrin. In this study, the authors found values between 144.87 and 396.09 μmol Fe^2+^ g^−1^. The antioxidant activity by ABTS and DPPH for MECP were lower than the antioxidant activity found in the different propolis encapsulated by [30] but higher than the results obtained by [1]. The authors of [37,42] inferred a relationship between phenolic compounds present between lyophilized and microencapsulated propolis extracts.

In agreement with our previous report [27] with microencapsulated grape pomace, the microencapsulation favored a reduction in the content of bioactive compounds with antioxidant activity when compared to lyophilized extracts. In fact, the spray drying process favored the degradation of bioactive compounds, which depends on the analyzed matrix and processing conditions. In this case, the oxidation reactions may have occurred during microencapsulation when the antioxidant activity of the bioactive compounds was reduced. This result corroborates the information described in [1], which addressed the possibility of using microencapsulated material in foods as a source of phenolic compounds, though with more modest antioxidant activity.

### 3.2. Lipid Oxidation and Fatty Acid Profile of Starch Biscuits

Foods prepared with oils and fats are subject to lipid oxidation, leading to undesirable substances that can compromise the quality of food. In general, there are no reports in the literature [46,47,48] addressing lipid oxidation by the TBARS method in starch biscuits (polvilho biscuit) based on canola oil, which is an aspect of the present study. Most studies have focused on meat products [49,50]. It is possible to find studies of lipid oxidation in cookies and biscuits [47,51], but there are no parameters that define the degree of oxidation by the TBARS method for baked products [47]. The authors of [46] analyzed two formulations of bread that differed in vegetable oil and antioxidants. In this study, the bread formulated with canola oil showed decreased MDA behaviour after six days of storage, obtaining values similar to that observed with starch biscuits with 11.75 mg of MDA kg^−1^ of sample.

The results of the TBARS analysis for the four starch biscuits formulations are presented in Figure 2.

The TBARS values observed in starch biscuits during the analysis period ranged from 12.87 to 6.01 mg MDA Kg^−1^ of biscuit; values higher than 10 mg MDA kg^−1^ of biscuit were observed only on the day of processing. Even though there are no statistically significant differences between the different formulations, a small decrease was seen in malonaldehyde content in all formulations, suggesting that the lipid oxidation process reduced with storage time. In the analysis of starch biscuit, it was predicted that the effect of the application of antioxidants would cause significant differences between the control samples and those in which the antioxidants had been applied. However, in the same way as [52], in studies of accelerated lipid oxidation tests on canola oil treated with pitanga leaf extracts, it was found that the extracts’ effect was close to those observed in samples with BHT.

Although the MDA values for F3 (MECP) at the beginning of the experiment were not the lowest, at the end of 45 days of storage, the MECP (F3) biscuits showed the most significant reduction with an MDA content of 48.88% compared to the values obtained on the day of processing. The biscuits containing LFCP extract (F4) showed a 44.15% reduction in MDA content between zero and 45 days of storage. At the end of the storage time, there was no significant difference between the biscuits containing MECP, LFCP, and BHT. Although MDA values for these samples were similar, the adverse health effects associated with BHT use in food, and the fact that it has been banned in some countries, justify using natural antioxidants of the co-product of propolis LFCP and MECP.

The four formulations of starch biscuit did not present significant variation among themselves regarding the total lipid content, with obtained values of 12 to 15% of lipids, as shown in Figure 3.

In this study, it was possible to identify and quantify in the formulations of starch biscuit the following fatty acids: palmitic acid (C16:0), stearic acid (C18:0), oleic acid (C18:1n-9), linoelaidic acid (trans C18:2), α-linoleic acid (C18:2n-3), and C18:2n-6 (linoleic acid). The variations in the concentrations of fatty acids identified in starch biscuits, expressed in g 100 g^−1^ of the sample, are presented in Figure 4.

The fatty acids identified in the biscuit samples corresponded to the fatty acids characteristic of canola oil and observed in previous studies of this same lipid source [52,53,54,55].

The concentrations of oleic, linoelaidic, linoleic, and linolenic acids in starch biscuit did not present significant variations between the different formulations during the storage period. Palmitic (C16:0) and stearic (C18:0) saturated fatty acids demonstrate a more pronounced variation on the day of processing and up to fifteen days of storage, with emphasis on the variation observed for C16:0 starch biscuits with BHT (F2). However, from 30 and 45 days of storage, the formulations showed higher constancy in the content of saturated fatty acids C16:0 and C18:0 (Figure 4).

The behaviour of the total lipids, saturated (SFA), monounsaturated (MUFA) and polyunsaturated (PUFA) fatty acids, as well as the relationship between PUFA/SFA in the starch biscuit formulations during the 45 days of storage, are shown in Figure 5.

The relationship between these groups of fatty acids contributes to understanding the variation of lipid composition [53,54,55,56].

Except at 15 days storage, F2 presented a lower rate/ratio PUFA/SFA, possibly due to the outstanding positive variation in C16:0 concentration. In the other periods, the ratio (PUFA/SFA), even if subtle, was always higher for the different formulations according to the amount of antioxidant employed (mg/g of fat); that is, in 0, 30, and 45 days of storage the PUFA/SFA ratio followed the behaviour: F4 > F3 > F2 > F1. The maintenance of the fatty acid profile, especially the PUFA contents, is relevant given the benefits provided by these fatty acids to human health.

The observations taken concerning the composition and behaviour of the fatty acids in starch biscuit do not represent the absence of an expected effect when applying the natural antioxidants MECP and LFCP. On the contrary, these had a positive impact on fatty acid composition and behaviour. The analysis of fatty acids during the storage period did not show alterations that would be characteristic of a lipid oxidation process; in this sense, the analysis demonstrated the positive effect of the antioxidant application on oxidative stability.

## 4. Conclusions

LFCP presented antioxidant activity higher than the values achieved by MECP, even with a microencapsulation process efficiency of 63%. The evaluation of lipid oxidation by the TBARS test in starch biscuits based on canola oil was considered and can serve as a reference for new foods with the same characteristics. The profile and concentrations of fatty acids by GC-FID of the starch biscuit samples refer to the lipid matrix (canola oil). The application of the natural antioxidants MECP and LFCP demonstrated positive effects on the maintenance of fatty acids during the observation period, showing strong potential as ingredients in the formulation of foods to prevent lipid oxidation in starch biscuits.

## Figures and Tables

**Figure 1 molecules-26-06400-f001:**
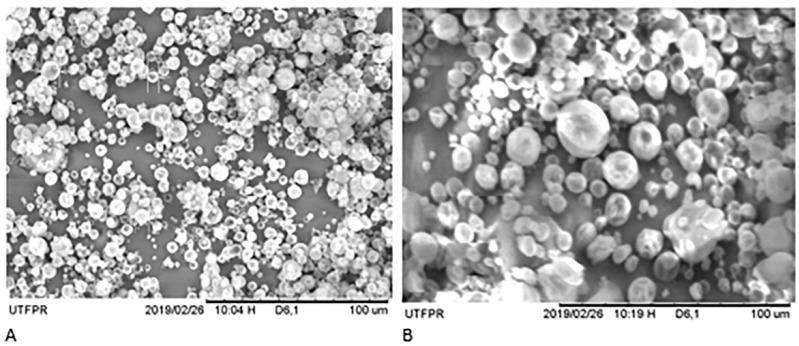
Microencapsulated extract of the propolis co-product. Magnifications: (**A**): 800×; (**B**): 1000×.

**Figure 2 molecules-26-06400-f002:**
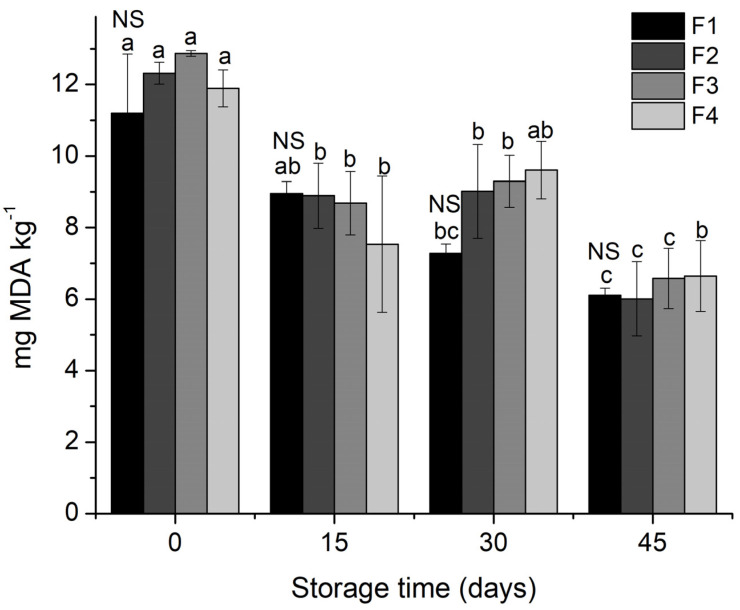
Average values of TBARS (mg of MDA Kg^−1^) in starch biscuit formulation during storage time. MDA: malonaldehyde; F1: starch biscuit control (no added antioxidant); F2: starch biscuit with butylated hydroxytoluene (BHT); F3: starch biscuit added microencapsulated propolis co-product (MECP); F4: starch biscuit added lyophilized propolis co-product (LFCP). NS: there are no statistical differences (*p* > 0.05) by the Tukey test between the formulations in the respective storage time. Different lower-case letters above the formulation column represent statistical differences (*p* < 0.05) by the Tukey test for the formulation over the storage time.

**Figure 3 molecules-26-06400-f003:**
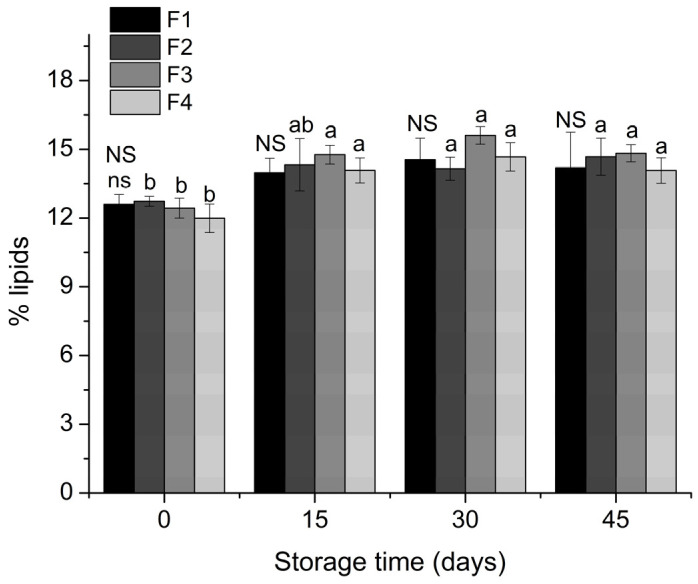
% Total lipids for the four starch biscuit formulations during storage time. F1: starch biscuit control (no added antioxidant); F2: starch biscuit added butylated hydroxytoluene (BHT); F3: starch biscuit added microencapsulated propolis co-product (MECP); F4: starch biscuit added lyophilized propolis co-product (LFCP). NS: there are no statistical differences (*p* > 0.05) by the Tukey test between the formulations in the respective storage time. Different lower-case letters above the formulation column represent statistical differences (*p* < 0.05) by the Tukey test for the formulation over the storage time.

**Figure 4 molecules-26-06400-f004:**
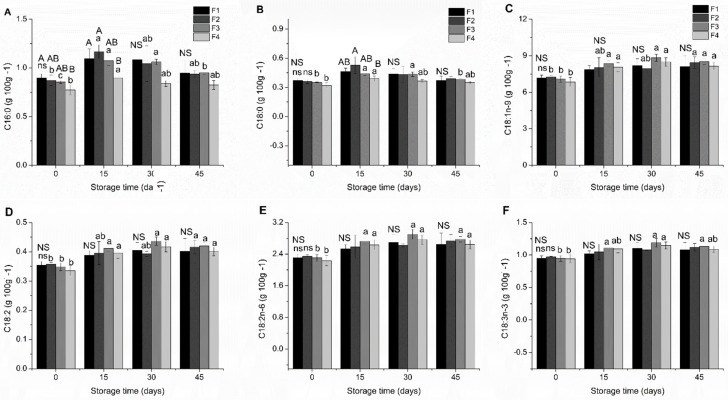
Fatty acids identified in starch biscuit formulation during storage time. (**A**) palmitic acid, (**B**) stearic acid, (**C**) oleic acid, (**D**): linoelaidic acid; (**E**) linoleic acid and (**F**) α-linolenic acid. F1: starch biscuit control (no added antioxidant); F2: starch biscuit added butylated hydroxytoluene (BHT); F3: starch biscuit added microencapsulated propolis co-product (MECP); F4: starch biscuit added lyophilized propolis co-product (LFCP). NS: there are no statistical differences (*p* > 0.05) by the Tukey test between the formulations in the respective storage times. Different capital letters above a column of information represent statistical differences (*p* < 0.05) by the Tukey test between the formulations in the respective storage times. Different lower-case letters above the column of the information column represent statistical differences (*p* < 0.05) by the Tukey test for the formation during the storage time.

**Figure 5 molecules-26-06400-f005:**
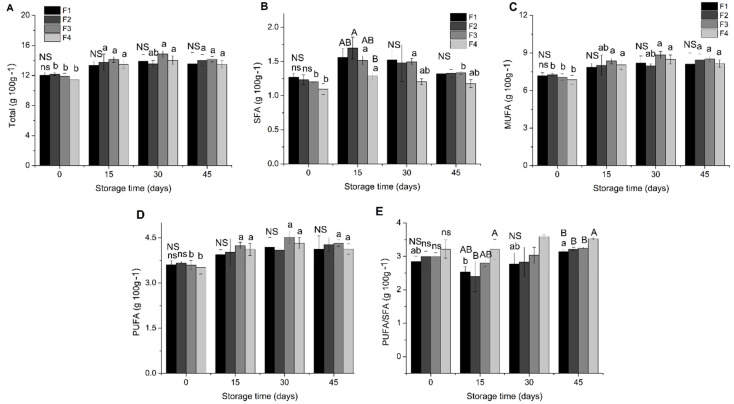
Fatty acids of starch biscuit formulation during storage. (**A**) Total FA: fatty acids; (**B**) SFA: saturated fatty acids; (**C**) MUFA: monounsaturated fatty acids; (**D**) PUFA: polyunsaturated fatty acids; (**E**) PUFA/SFA. F1: starch biscuit control (no added antioxidant); F2: starch biscuit added butylated hydroxytoluene (BHT); F3: starch biscuit added microencapsulated propolis co-product (MECP); F4: starch biscuit added lyophilized propolis co-product (LFCP). NS: there is no statistical difference (*p* > 0.05) by the Tukey test between the formulations in the respective storage time. Different capital letters above a column represent statistical difference (*p* < 0.05) by the Tukey test between the formulations in the respective storage time. Different lower-case letters above the column represent statistical difference (*p* < 0.05) by the Tukey test for the different storage time.

**Table 1 molecules-26-06400-t001:** Total phenolic compounds and antioxidant activity.

Parameters	LFCP	MECP
TPC (mg GAE g^−1^)	199.78 ± 0.28	69.28 ± 0.33
DPPH (μmol Trolox g^−1^)	496.28 ± 0.00	47.02 ± 0.00 *
ABTS (μmol Trolox g^−1^)	5041.81 ± 0.00	485.92 ± 0.01 *
FRAP (μmol Fe^2+^ g^−1^)	3796.28 ± 0.00	386.69 ± 0.01 *

TPC: total phenolic compounds; GAE: gallic acid equivalent; LFCP: lyophilized propolis co-product; MECP: microencapsulated propolis co-product. * represents significant difference by the *t* test (≤0.05).

## Data Availability

The data presented in this study are available on request from the corresponding author.

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
