# Peer review of "Microencapsulated and Lyophilized Propolis Co-Product Extract as Antioxidant Synthetic Replacer on Traditional Brazilian Starch Biscuit"

_molecules, 2021, doi:10.3390/molecules26216400_

Round 1

Reviewer 1 Report

The manuscript entitled “Microencapsulated and Lyophilized Propolis Co-Product Extract as Antioxidant Synthetic Replacer on Traditional Brazilian Starch Biscuit” aims to determine the propolis co-product extracts capability to reduce the lipid oxidation in starch biscuit formulated with canola oil. The authors synthesized two types of antioxidant replacer: microencapsulated propolis co-product (MECP) and lyophilized propolis co-product (LFCP) and analyzed their total phenolic content and antioxidant activity. As I said before, the idea is very good, and the topic is interesting and significant. The experiments were conducted systematically, and the results are promising. Still, the improvements made were not extensive. Although the Introduction and Methods are improved, I do not see much more alterations. I still think that the presentation of the results should be written more clearly, since in the section Results and discussion almost are no changes. It would be a shame not to invest more time to make this manuscript as good as the results obtained.

Author Response

Thank you very much for inviting us to submit a revised version of our manuscript entitled: "Microencapsulated and Lyophilized Propolis Co-Product Extract as Antioxidant Synthetic Replacer on Traditional Brazilian Starch Biscuit" for publication in Molecules, the reviewers have raised important concerns. We have now made a revision of the manuscript taking into account these points, which helped in improving the quality of the manuscript. We have provided a point-by-point response to all the comments.

Response: Thank you for the reviewers' considerations. In this version more discussion about the microencapsulation process and antioxidant activities have been included. An Engglish language revision was made.

Reviewer 2 Report

The authors have satisfactorily responded all my questions

Author Response

 The authors have satisfactorily responded all my questions.

Response: Thank you for making it possible to improve the article.

Reviewer 3 Report

The manuscript entitled "Microencapsulated and lyophilized propolis co-product as antioxidant synthetic replacer on traditional Brazilian starch biscuit" may be considered for publication after addressing the following recommendations:

Line 49-51: Have the propolis encapsulation studied before by other authors?

Line 85: the exact encapsulant has to be specified.

Lines 111-117: The protocol of the emulsion preparation is not complete. More data is required so that another researcher can repeat the emulsion preparation.

Line 246: standard deviation of the encapsulation efficiency is required.

Line 256: Particles do not present cylindrical shape. Please correct.

Table 1: The results presented in this table are surprising, because according to the authors an order of magnitude in those properties is lost because of the heat of the spray drying. Is it possible that the calculations are made with respect to the total amount of encapsulated microparticles and not with respect to the dried extract? Please, verify.

Author Response

Thank you for making it possible to improve the article.

Line 49-51: Have the propolis encapsulation studied before by other authors?

Yes, propolis encapsulation was studied by other authors, such as Jansen-Alves et al. (2018), Spinelli et al. (2015), Da Silva et al. (2013).

Jansen-Alves, C.; Fernandes, K.F.; Crizel-Cardozo, M.M.; Krumreich, F.D.; Borges, C.D.; Zambiazi, R.C. Microencapsulation of propolis in protein matrix using spray drying for application in food systems. Food and Bioprocess Technology 2018, 11(7), 1422–1436. https://doi.org/10.1007/s11947-018-2115-

Spinelli, S.; Conte, A.; Lecce, L.; Incoronato, A. L.; Del Nobile, M.A. Microencapsulated propolis to enhance the antioxidant properties of fresh fish burgers. Journal of Food Process Engineering 2015 38(6). https://doi.org/10.1111/jfpe.12183

Da Silva, F.C.; Da Fonseca, C.R.; De Alencar, S.M.; Thomazini, M.; Balieiro, J. C.D.C.; Pittia, P.; Favaro-Trindade, C.S. Assessment of production efficiency, physicochemical properties and storage stability of spray-dried propolis, a natural food additive, using gum Arabic and OSA starch-based carrier systems. Food and Bioproducts Processing 2013, 91(1), 28–36. https://doi.org/10.1016/j.fbp.2012.08.006

Line 85: the exact encapsulant has to be specified.

Sorry for our mistake. More complete information was entered since the encapsulating agent used was maltodextrin.

Lines 111-117: The protocol of the emulsion preparation is not complete. More data is required so that another researcher can repeat the emulsion preparation.

We appreciate your attention and add clearer information in the methodology and also add references.

Line 246: standard deviation of the encapsulation efficiency is required.

Sorry for our mistake. The test was carried in triplicate and the standard deviation was mentioned in the text.

Line 256: Particles do not present cylindrical shape. Please correct.

We agree with the reviewer and this error has been fixed. The microparticles had a spherical shape.

Table 1: The results presented in this table are surprising, because according to the authors an order of magnitude in those properties is lost because of the heat of the spray drying. Is it possible that the calculations are made with respect to the total amount of encapsulated microparticles and not with respect to the dried extract? Please, verify.

Yes, in fact, based on the findings, microencapsulation favoured a reduction in the content of bioactive compounds with antioxidant activity when compared to lyophilized extracts. We believe that lyophilization was an excellent technique for maintaining phenolic compounds, while the spray drying process favoured a degradation of bioactive compounds, which depend on the analyzed matrix and processing conditions. Furthermore, the variation in the antioxidant activity of lyophilized and microencapsulated products may be related to different interactions between the sample and the carrier agent, responding in different ways and often increasing or decreasing their activity. In our case, what is known is that oxidation reactions may occur during microencapsulation and a part of the bioactive compounds may have had their antioxidant activity reduced. A new discussion about it was added.

Reviewer 4 Report

The authors have implemented the manuscript in this round of revision by adding more information on the total phenolic contents of the extracts. I think that taken together, the results are valuable for publication in the context of the food science and technology field, although this manuscript lacks a comprehensive characterization of the phenolic profile of the extracts as assessed by LC-MS or LC-MSMS. 

Author Response

Thanks for the suggestion. We agree with the reviewer regarding the use of more sophisticated techniques to analyze the profile of phenolic compounds, however, as we did not have the equipment available, we tried to focus on the oxidation of fatty acids. New studies can be conducted from this and we can include new partnerships to be able to carry out these analyses.

Round 2

Reviewer 1 Report

I recommend this manuscript for publication. 

Author Response

Dear

Thank you for giving us the opportunity to improve the manuscript.
Reviewer 1:
Thank you for the reviewers' considerations. A language review was carried out.

Reviewer 3 Report

I think it would be interesting to include previous work done in propolis encapsulation in the introduction.  By doing this last improvement, I think that the manuscript is ready for publication

Author Response

Dear

Thanks for the suggestion, and was included other information in the Introduction, with more references about the theme. Please see the new version of the manuscript.

This manuscript is a resubmission of an earlier submission. The following is a list of the peer review reports and author responses from that submission.

Round 1

Reviewer 1 Report

The manuscript entitled “Microencapsulated and Lyophilized Propolis Co-Product as Antioxidant Synthetic Replacer on Traditional Brazilian Starch Biscuit” aims to determine the propolis co-product extracts capability to reduce the lipid oxidation in starch biscuit formulated with canola oil. The authors synthesized two types of antioxidant replacer: microencapsulated propolis co-product (MECP) and lyophilized propolis co-product (LFCP) and analyzed their total phenolic content and antioxidant activity. The idea is very good, and the topic is interesting and significant. The experiments were conducted systematically, and the results are promising. Still, the manuscript is written kind of blurry, and it should be improved. The presentation of the results should be written more clearly. The language should be improved. It would be interesting to use MECP and LFCP together and test if there is some synergistic effect.

Reviewer 2 Report

The authors in the paper describes a possibility usage of propolis co-product extracts' in foods as antioxidant agents. The paper is interesting but it presents some flaws I hope my comments will be seen as constructive and beneficiary to the discussion

The author must follow the journal guidelines while writing the manuscript: see references

Please insert the SEM imagines of raw material (propolis and maltodextrin) before microencapsulation process and the lyophilized co-product propolis.

Please improve the magnifications and the focus of SEM imagines.

Please insert in the all antioxidant test the positive controls

Reviewer 3 Report

The manuscript entitled "Microencapsulated and lyophilized propolis co-product as antioxidant synthetic replacer on traditional Brazilian starch biscuit" may be considered for publication after addressing the following recommendations:

Title

I think it is required to include the word extract in the title to avoid misunderstandings.

Abstract

I think it is necessary to clarify that an extract of the propolis co-product was used.

The used encapsulating material is not mentioned.

Keywords

I think it is required to add the antioxidant keyword to the list, and substitute propolis co-product by ethanolic extract of propolis co-product.

Introduction

Provides information about the state of the art on propolis. However, I think it should be rewritten to reorganize the ideas and make it easier to read.

In Line 41, authors mention that the encapsulation could be performed by spray drying, but later in the article is also performed by lyophilization. Probably, it would be interesting to include a few lines in the introduction explaining this decision.

What encapsulating material was used?

Materials and methods

Line 80: How was the extract concentration determined?

Line 82-87: Was an encapsulating material used for the lyophilization?

Lines 88-91: More details should be provided about the spray drying procedure. What encapsulating material was used? In which concentration? What were the parameters of the spray drying? What is the device used?

Lines 92-98: More details should be provided on the emulsion preparation: Why do the authors increase temperature to dissolve maltodextrin? How the authors know that they are preparing an emulsion if both phases (water – ethanol) are miscible? Which ratio of dispersed phase – continuous phase was used? Was a surfactant used to stabilize the emulsion? Which was the emulsion droplet size? And its stability?

Lines 107-111: Even if the procedure was similar to previous publications, I consider it necessary to provide a summary of the procedure.

Lines 115-121: More information should be provided on microscopy analysis. Were the samples sputtered previous to SEM? Was the lyophilized sample also analyzed? Was an analysis of the particle size performed?

Lines 122-125: How much sample was used?

Line 127: This sentence has to be rewritten to be readable.

Line 132: Data about the spectrophotometer should be included.

Lines135-138: A summary of the procedure should be provided.

Lines 139-146: Which amount of sample was used? At which concentration? How much ABTS radical was used?

Lines 147-151: Which concentration was used of the aliquots of the samples? Could you provide a summary of the procedure?

Lines 160-162: What was the criteria followed to decide the amounts of antioxidants?

Lines 167-174: Was the pure canola oil analyzed? Could you provide summary of the procedure? Which amount of sample was used? Which amount of reactives?

Lines 175-188: Could you provide a summary of the procedure for the esterification? How was the biscuit sample prepared for GC?

Results and discussion

Line 204: Is 63% an average value? Could you provide also the standard deviation? Which could be the reason to obtain only a 63% of encapsulation efficiency?

Lines 212-214: What is the relationship between the geographic origin and seasonality, sample preparation and extraction method with the encapsulation efficiency? These factors could affect the loading capacity, but I do not see the point with the encapsulation efficiency.

Line 214-215: Referring to the microparticles morphology, do the authors mean spherical instead of cylindrical? Do you have better SEM images? Do you have SEM images for the lyophilized sample? What is the size distribution of these microparticles? What is the cause of the depressions observed on the microparticles surface?

Lines 224-229: What is the reason for such a big difference between the results obtained for spray drying and lyophilization? Were the results corrected by the amount of extract in each type of particle?

Lines 258-260: This sentence has to be checked and rewritten.

Lines 284-293: According to Figure 2, it seems that the spray drying encapsulation was not very effective in terms of protection. What could be the reason?

Reviewer 4 Report

This is an interesting manuscript dealing with food science and technology considering that the authors evaluated the utilization of microencapsulation to preserve the antioxidant activity of a propolis residue to be added to starch biscuits in order to avoid and slowing down lipid oxidation. The manuscript is well prepared and the authors found that the spray drying technology is able to provide an efficiency of 63% regarding its microencapsulation ability. Overall, I think that the manuscript should be revised when considering the English style and discussion of the main findings. The Introduction section is too short and few information is provided to the reader. Please add more references regarding the microencapsulation of bioactive compounds in food technology by assessing also drawbacks and advantages. Also, any hypothesis regarding the impact of the lyophilization process on some bioactive compounds of your extracts? One of the main drawbacks of this paper is the utilization of an in vitro spectrophotometric assay to assess the phenolic content (likely one of the main contributors to the antioxidant activity of the microencapsulated extracts). I would like to see more information as provided by HPLC-UV or HPLC-MS or HPLC-MSMS.